# Boosting the Full Potential of PyMOL with Structural Biology Plugins

**DOI:** 10.3390/biom12121764

**Published:** 2022-11-27

**Authors:** Serena Rosignoli, Alessandro Paiardini

**Affiliations:** Department of Biochemical Sciences “A. Rossi Fanelli”, Sapienza Università di Roma, 00185 Rome, Italy

**Keywords:** PyMOL, bioinformatics, plugin, molecular viewer, structural biology, sequence analysis, molecular docking, molecular dynamics, structure-function analysis, protein structure prediction, virtual screening

## Abstract

Over the past few decades, the number of available structural bioinformatics pipelines, libraries, plugins, web resources and software has increased exponentially and become accessible to the broad realm of life scientists. This expansion has shaped the field as a tangled network of methods, algorithms and user interfaces. In recent years PyMOL, widely used software for biomolecules visualization and analysis, has started to play a key role in providing an open platform for the successful implementation of expert knowledge into an easy-to-use molecular graphics tool. This review outlines the plugins and features that make PyMOL an eligible environment for supporting structural bioinformatics analyses.

## 1. Introduction

The first version of the PyMOL graphical environment for modeling and visualization of molecules, which dates back to 2000, was developed by Warren Lyford DeLano and was at first distributed by DeLano Scientific LLC [1]. After the unexpected death of Warren DeLano in 2009, PyMOL was acquired by Schrödinger Inc. (NY, USA) [2], which continued to support the open-source vision of the original author, but also provided commercial support for maintenance and access to additional features. Currently, PyMOL is a cross-platform, open-source but proprietary software program, maintained, developed and supported by Schrödinger Inc., which also reserves licensing rights. The major changes since Schrödinger Inc. acquisition released with PyMOL version 2 (PyMOL 2), feature a new Graphical User Interface (GUI) that makes use of PyQt5 [3], and replaced Tcl/Tk [4]. PyMOL 2 packages and dependencies management are now on Anaconda [5], since version 2.3, PyMOL started to be developed and released in Python 3 [6]. Such novelties made it possible for PyMOL to be continuously up to date, arriving at the latest available version 2.5, which was released in October 2021. The use of efficient and high-performing graphical libraries, i.e., Open Graphics Library (OpenGL) [7] and the GL Shading Language (GLSL), underlies the advanced images and light renderings of PyMOL, which in turn permits deep control over the setup of final image appearance [8]. Indeed, PyMOL is widely exploited by the scientific community for creating high-quality images and videos that accurately depict the molecular structure being presented [9,10].

One of the main features of PyMOL is the ease for external developers to boost its functionality via scripts and plugins. In this way, PyMOL is programmatically accessible through its editable command-line and Application Programming Interface (API), which make PyMOL features easily exploitable by external software. As a consequence, sequence analysis, molecular docking, molecular dynamics, structure-function relationships analysis, protein structure prediction and virtual screening are all well-known structural bioinformatics approaches for which PyMOL plugins have been developed over the years [11,12]. In addition to plugins, some useful tools are inherently implemented in PyMOL. For example, PyMOL handles molecular building and editing by providing a builder menu to draw small chemical compounds, peptides and nucleic-acids sequences. The builder menu comes with complete hydrogen and charge fixing tools and with 3D-model cleaning strategies. In the same context, PyMOL supports the possibility to mutate a protein residue or a nucleotide base with the ‘Mutagenesis Wizard’. The ‘Sculpting Wizard’ gives access to a real-time energy minimization function implemented in PyMOL, which is especially useful when dealing with chemical editing.

PyMOL is an ideal candidate for developing advanced Computational Protein Analysis (CPA) pipelines. Its two-pronged nature of molecular graphics viewer and programmatically accessible interface make it particularly suited to easily develop Python-based software tools for advanced investigations of biological macromolecules. A paper in 2017 outlined PyMOL plugins and features that are useful in computational drug design [12]. Some of these plugins are still actively maintained and are worth citing: CAVER 3.0, a tool for the analysis of transport pathways in dynamic protein structures [13]; PYROSETTA, an interactive interface to the powerful Rosetta molecular modeling suite [14]; Dynamics, which adds Gromacs-based molecular dynamics simulation features to PyMOL [15], and the APBS Electrostatics plugin, which implements the Adaptive Poisson–Boltzmann Solver [16] is suited for the analysis of solvation and electrostatics in protein structures. The latter, which was has been reviewed in [12] as a standalone plugin, is now available along with PyMOL installation. In addition to those that were previously available, several other PyMOL plugins have been released in recent years. The main aim of this review is to provide an overview on the newly released PyMOL plugins, which are summarized in Table 1, and their scope in CPA. These are introduced by dividing the CPA realm into three categories: (i) Protein Sequences and Structures Analyses (PSSAs); (ii) Protein-Ligand Interactions (PLI); (iii) Protein Dynamics (PD).

## 2. Protein Sequences and Structures Analyses (PSSAs)

PSSAs are crucial in a variety of biological research areas, particularly in the prediction and modeling of protein structures, which include similarity searches, alignments of sequences and structures, evolutionary and structural comparison, and homology modeling. PyMOL is widely used for PSSAs due to the visual and editing support that is provided for the sequences (i.e., protein, RNA and DNA sequences) of the loaded associated 3D structures.

### 2.1. PyMod

PSSAs include a wide range of often-combined methods such as similarity searches, alignments of sequences and structures, evolutionary comparisons and homology modeling. For example, the template-based prediction of protein 3D quaternary structures in complex with ligands and other heteroatoms requires: (i) databases search for homologous protein (s) with known 3D structure (s); (ii) multiple sequence and structure alignments tools, and (iii) algorithms for protein structure prediction and quality assessment. The PyMod plugin [17] provides a GUI-based and easy-to-use environment for PSSAs by implementing, in an integrated manner, a wide range of algorithms, e.g., (PSI-)Blast [18], MUSCLE [19], ClustalW [20], Clustal Omega [21], WebLogo 3 [22], ESPript 3.0 [23], CAMPO, SCR-Find [24], PsiPred [25] and MODELLER modules for 3D model building [26], sequence-structure alignment [27] and quality assessment [28]. 

The latest version of PyMod, i.e., PyMod 3, runs on Windows, MacOS, and Linux on both the open source and incentive PyMOL builds provided by Schrödinger Inc. Alignments, sequences or structures can be imported in PyMod, e.g., protein structures that are loaded in PyMOL. Any imported object is processed by PyMod to retrieve the associated information, e.g., heteroatoms and water molecules, and to display its sequence in the dedicated window. The latter provides the interface for the many PyMod functionalities, and is designed to visually support manual editing of sequences and alignments. The plugin so constituted is widely used for template-based protein structure predictions (Figure 1a–d), alignments, evolutionary conservation analyses, building of phylogenetic trees and database searches [29,30,31,32]. Protein structure analyses now often rely on AlphaFold (AF) [33] accurate predictions; nonetheless, some underlying dynamics features of proteins may be unveiled by comparing experimental and AF-predicted models. Such a hybrid approach was applied in a recent study on the human trans-3-hydroxy-L-proline dehydratase [34], in which the PyMod support for structural analyses, e.g., structure divergence plots, facilitated the comparisons between experimental and predicted 3D protein structures in different conformational states.

### 2.2. pyProGA

Protein residue network (PRN) methods model proteins according to the graph theory [37] to obtain a reduction of the complexity of protein structures into more simple descriptors (residues as vertices/nodes and the interactions between them as edges) [38]. Analyses such as communication pathways, allostery effects and networks of interactions can all be addressed by making use of PRN-based analyses [39,40,41,42]. The plugin Protein Graph Analyser (pyProGA) [43] offers a GUI for several PRN methods, i.e., centrality calculation, graph partitions evaluation, shortest path identification, ego graph, network differential analysis, binding energy computing, singular value decomposition technique and pair interaction energies analysis. Graphs solely computed from structural data (PDB file format) are distance-based networks (D-PRN). Otherwise, if an output from either the fragment molecular orbital [44] or an Amber calculation [45] (“.prmtop” file) is provided, the computed graphs are based on residue-pair interaction energies (PIE-PRN). The results can be inspected directly in the pyProGA window and/or in PyMOL (Figure 1e). The pyProGA window accommodates plot visualization, while PyMOL is used for color mapping the results on the analyzed proteins (Figure 1f,g). A PRN-based metric, the residue folding degree, has been proposed as a means to measure the propensity of the protein backbone to form secondary structures in Molecular Dynamics (MD) simulations [46,47]. The residue folding degree analysis relies on conformational state descriptors, namely subgraph centrality, which can be computed from the pyProGA plugin. Additional metrics computable by pyProGA are appropriate for the identification of residues mediating key interactions between proteins, as showed with the de novo synthesized TIM barrel protein (PDB-ID: 5BVL) [48], for which singular value decomposition (SVD) analysis was applied to identify protein-protein interactions between monomers [43].

### 2.3. MPBuilder

Membrane protein 3D structures are hard to solve, due to their inherent association with lipidic macromolecules that makes the protein solubilization a crucial point [49,50]. Despite the challenges, many protocols have been developed for solving accurate 3D structures of membrane proteins, but are still cumbersome, expensive and time-consuming [51]. The Small-angle X-ray scattering (SAXS) technique is preferred because of less stringent sample preparation requirements. The output of a SAXS experiment needs to be interpreted according to computational modeling approaches [52,53]. A PyMOL plugin, MPBuilder [54], has been developed for assisting the modeling of membrane proteins on SAXS-derived data. Two tabs, for building and refinement (Figure 1j–k), guide the user through the analysis. The required inputs are a file storing SAXS data and the choice of an appropriate protein-lipidic assembly to guide the model building. MPBuilder greatly exploits PyMOL embedded functions, such as translation, rotation, atomic distance computation and atoms selection, as well as protein and lipidic visualization (Figure 1l). The implementation in the PyMOL framework makes it accessible for regular use, even by non-experts. For instance, MPBuilder was used for the interpretation of SAXS data of the phosphoethanolamine enzyme that is active at the lipidic interface, namely lipid A of pathogenic gram-negative bacteria as an antibiotic resistance mechanism [55].

### 2.4. ProBiS H_2_O, ProBiS H_2_O MD and Waterdock 2.0

Water solvation is well known for its influence in protein conformation, pharmacodynamics, rational drug design and protein structure prediction [56,57]. Given the relevance of water-mediated interactions, a PyMOL plugin called ProBiS H_2_O has been developed to infer conserved water binding sites by exploiting information derived from experimentally solved water-containing structures [58]. Compared to other computationally expensive methods, such as molecular dynamics-based methods, it provides an easy and fast way for conserved water site identification. The PDB-ID of the protein of interest is the unique mandatory input of ProBiS H_2_O, starting from which similar proteins are retrieved, while taking into consideration a user-defined percentage of sequence identity. Afterwards, local superimposition, sampling and clustering of water molecules are carried out and a conservation score is computed for each cluster. In PyMOL, water molecules are visualized and colored according to the computed conservation score. Recently, a new version of ProBiS H_2_O was released, called ProBiS H2O MD [59]. To account for lesser known proteins, hence proteins lacking enough experimental data for using the standard ProBiS H_2_O pipeline, this new version of the plugin optimizes the algorithm extending experimental data with MD trajectories. The latter must be a file (i.e., a “.dcd” file) derived from MD simulations of the protein of interest in a water box. At the end of the analysis, from within the plugin window, alternative visualizations of the clusters can be chosen and the amino acids encompassing the chosen binding site can be visualized (Figure 1h,i). An interesting application of ProBiS H2O MD elucidated the influence of local hydration on the PP1-Src kinase ligand binding [60]. Alternatively, interacting water molecules can be inferred by making use of molecular docking engines, as it is done by Waterdock 2.0 [61]. This tool implements Autodock Vina and uses optimized parameters to dock water molecules in protein structures (Figure 2a–c). The influence of water-mediated interactions on protein-ligand complexes was recently investigated by making use of Waterdock 2.0 (in comparison with other software intended for the same aim), for estimating the binding free-energy of human acetylcholinesterase in a complex with tacrine at different solvation states [62].

### 2.5. iPBAvizu

The functional and structural characterization of protein folding and conformational variability often employs structure superposition and comparison. PyMOL’s inherent alignment method is sequence-based, i.e., it makes use of a sequence alignment followed by a structural superimposition. However, for an efficient structure superimposition, related proteins with very low sequence identity must rely on structure-based alignment algorithms such as DALI [63], CE [64], SSAP [65] and iPBA [66]. The iPBAVizu plugin [67] implements iPBA and simply requires a minimum of two 3D protein structures loaded in the PyMOL workspace to compute the structural alignment. The inferred sequence alignment is then visualized in the plugin window and the 3D structures are superimposed in PyMOL (Figure 2d). Many analyses [68,69] are now relying on iPBAVizu for efficient protein structure-based superimposition. For instance, iPBAVizu guided the structural comparison of the primary contributors to SARS-CoV-2 antigenicity for the in-silico design of a multi-epitope vaccine candidate [70].

### 2.6. DCA-MOL

Multiple Sequence Alignments (MSAs) have been used for decades to obtain structural, functional and evolutionary information on proteins [71,72]. MSA-based analyses have applications in structure prediction, protein conformation dynamics, analysis of folding pathways, identification of binding sites, inference of interaction interfaces and prediction of interacting partners [73,74,75,76]. Some methods that fall into the group of Direct Coupling Analyses (DCA) [77,78] are used to depict the inherent evolutionary information of MSAs of homologous proteins. Co-evolutionary couplings among residue pairs are quantified in DCA with a score, called ‘Direct Information’ (DI), which is computed for each pair of residues. However, to obtain a comprehensive understanding of such analyses, the results must be processed and mapped on 3D structures. The PyMOL plugin DCA-MOL [79] interactively visualizes coevolutionary residue-residue interactions in contact maps and 3D structures. By providing DI-containing files (Figure 2e), an MSA “.fasta” file and the 3D structure of the protein of interest, DCA-MOL maps the contacts onto the 3D structures loaded in PyMOL workspace and shows contacts and distances maps in its dedicated window (Figure 2f). The interactions are captured either in short (<8.0 Å) or long distances (>8.0 Å) and, when mapped in the same chain (intra-chain interactions), can support protein structure prediction analyses. However, since DCA data of concatenated MSAs can be interpreted, DCA-MOL is not limited to intra-chain analyses, but also inter-chain interactions can be analyzed. On the other hand, by providing more 3D structures of the same protein, the mapping of the analyzed coupling residues can be easily switched by DCA-MOL from one structure to another. In the case of multiple structures, provided by either a MD-trajectory or an experimental source, such visualization can help in rationalizing conformational variabilities. These case studies are deeply investigated in the tutorials provided by the developers (https://dca-mol.cent.uw.edu.pl; accessed on 14 October 2022).

## 3. Protein-Ligand Interactions

Protein-ligand interactions, prediction and analyses, comprise a set of widely used methods in Structure-Based Drug Discovery (SBDD), i.e., Molecular Docking (MDo) algorithms, re-scoring metrics, post-processing analyses and Virtual Screening (VS). Driven by the need to have in-hand and easy-to-access environments for SBDD, PyMOL has become a platform for assisting PLI analyses.

### 3.1. DockingPie

The widespread use of protein-ligand docking has led to the development of a wide range of search algorithms, scoring functions, procedures and post-processing analysis tools [80] that create diverse strategies and parameters to be set up by researchers. Although many tools are available for accomplishing each individual task, a lack of integration slows down and hampers the process. A newly developed PyMOL plugin, DockingPie [81], addresses such limitations by providing an interoperable implementation of the many tools that are needed to carry out each step of a MDo process, from the preparation of input files to the analyses of the results. Currently, the docking engines supported by DockingPie are Smina [82], Vina [83], RxDock [84,85] and ADFR [86] (Figure 3a), as well as their peculiar protocols such as the assignment of side chain flexibility or the setup of pharmacophoric restraints. From within DockingPie, a user can: (i) compute RMSD; (ii) visualize the results by means of predicted affinity-based plots, and (iii) carry out consensus scoring analyses. The input required is any ligand or protein currently loaded in the PyMOL workspace and, for ease of use, DockingPie handles the installation of all external dependencies. How to use the many functionalities of DockingPie is described in different tutorials (available at https://github.com/paiardin/DockingPie/wiki/Tutorials; accessed on 14 October 2022). It has been demonstrated that combining the results of different docking programs with the application of a consensus scoring algorithm can increase the success rates in VS processes [87,88]. In this context, due to the implementation of different docking engines and consensus scoring metrics, DockingPie is a tool that lets the user collect and re-score the results in a single integrated environment (Figure 3b–d), which is particularly useful in VS campaigns.

### 3.2. DRUGpy

Hot-spots of interactions between proteins and ligands, namely protein regions with potential positive contribution to the ligand binding free energy, are usually inspected at the early stages of SBDD. The FTMap algorithm [89], which is available as a web server, probes the entire protein surface for detecting hot-spots of interactions, and returns the associated physico-chemical properties. However, the raw nature of FTMap output files prevents the addressing of biologically relevant conclusions. For these reasons, the DRUGpy plugin [90] has been developed to analyze FTMap output data (Figure 3g), and evaluate the level of druggability of the corresponding hot-spots of interaction. DRUGpy quickly locates and defines pockets that FTMap data predict would bind drug-like compounds with high or low affinity (druggable sites), and also displays how protein conformational flexibility affects the target’s druggability. If a protein-ligand complex of an identified hot-spot is available, and is superimposed on the hot-spot, DRUGpy can compute fractional overlap to guide rational ligand design (Figure 3h). DRUGpy was used for the comparative analyses of two homologous enzymes, widely validated as drug targets, trypanothione reductase and glutathione reductase [90]. DRUGpy fractional overlap analysis of the druggable hot spots of both enzymes added significant physico-chemical considerations towards the design of species-specific inhibitors.

**Figure 3 biomolecules-12-01764-f003:**
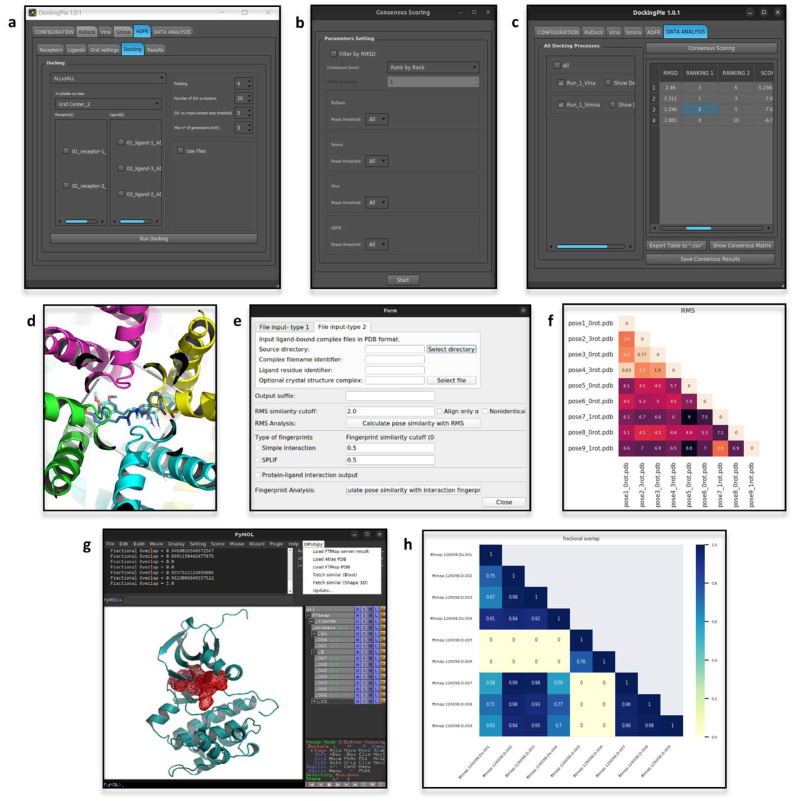
(**a**) DockingPie GUI (Windows OS) to set up an ‘all vs all’ MDo analysis with ADFR. (**b**) Appearance of the DockingPie windows (Ubuntu Linux OS) to run a consensus scoring analysis (**c**) and to inspect the results. In the latter, the re-scoring results are displayed in an interactive-table that can be clicked to directly visualize the molecules of interest in PyMOL (**d**). In this example, the re-scored poses were the results of two MDo runs carried out with Vina and Smina, from within DockingPie on a homo-tetrameric protein (PDB-ID: 5KMH [91]). (**e**) PoseFilter GUI to carry out either RMSD or interaction fingerprint comparison when a directory storing the objects to be analyzed as separated files (i.e., the protein and each conformation of the ligand) is provided. PoseFilter was used to analyze the MDo results obtained with DockingPie. (**f**) Heatmap of RMS values reporting the results of PoseFilter. (**g**) Visualization of the DRUGpy plugin and PyMOL workspace. The output data of a FTMap analysis on Aurora-A protein (PDB-ID: 1OL5 [36]) were loaded in DRUGpy for the analysis. DRUGpy automatically shows the identified hot-spots of interaction in PyMOL and, when a protein-ligand complex is analyzed, computes the fractional overlap analysis (shown as a heatmap (**h**)).

### 3.3. PoseFilter

MDo analyses, especially when carried out on a large-scale, return a huge number of binding conformations to be analyzed. This is very common for flexible, ensemble and consensus docking protocols, making necessary the use of a tool for post-processing analyses of docked conformations. Indeed, some MDo engines directly implement a filtering tool for discarding redundant poses. However, the fact that a lot of tools do not consider symmetric poses leads to a misinterpretation of molecule orientations and similarities. To overcome this issue, PoseFilter [92] implements two methods for post-processing analyses of docked complexes, i.e., root mean square deviations (RMSD) and interaction fingerprints, in a way that takes into account symmetric molecules. The correct interpretation of symmetric poses is a hurdle when ligands are docked in symmetrical binding sites, as usually occurs when a ligand binding pocket lies between adjacent monomers of oligomeric molecules. In analyzing these situations, PoseFilter gradually rotates the ligand and carries out an ‘all to all’ comparison, while discarding redundant poses. Either protein-ligand complexes or separated molecules can be provided (Figure 3e), and its use has been shown for (homo-) trimeric, dimeric and tetrameric proteins [92]. Since PyMOL is widely used for inspecting protein-ligand complexes, PoseFilter provides a quick and functional tool for analyzing MDo results (Figure 3f).

## 4. Protein Dynamics

The dynamic behavior of biomolecules is preferably studied with visual support. Indeed, many algorithms for PD have been implemented in different molecular graphics environments. Whereas the length and size of MD trajectories can hamper storing and visualization, PyMOL was adapted to handle the parsing and rendering of big MD-data files. Such novelty has become a step towards the use of PyMOL either for carrying out simulations or for the post-processing of PD-derived data.

### 4.1. Geo-Measures

The output of MD simulations is usually in the form of trajectory files that can be further interpreted, visualized and analyzed. Many python modules are available for molecular trajectory file interpretation, yet a free GUI for accessing such features is still missing. Geo-Measures [93] (Figure 4a) was born as a plugin for providing easy access to MDTraj [94], a python library of trajectory analyses modules. Given a topology and trajectory file (file formats supported by PyMOL are feasible), the following analyses can be carried out: (i) Cα and dihedral angle calculations; (ii) Cα triangle area; (iii) probability density function (PDF) of trajectory frequencies; (iv) root mean square deviation (RMSD) along trajectories; (v) radius of gyration (Rg); (vi) free energy landscape (FEL); (vii) principal component analysis (PCA); (viii) Ramachandran map; (ix) root mean square fluctuation (RMSF); (x) secondary structures definition; (xi) distance calculations, and (xii) trajectory paths visualization. Plots of the results can be visualized and exported as images from a dedicated window of Geo-Measures (Figure 4b–f) The authors provided practical examples of Geo-Measures use on hemoglobin and Ecto-5′-nucleotidase [93]. In a recent work on the mechanistic role of receptors-bound quercetin in hepatoprotection [95], the bound state stability, the compactness and conformational fluctuations were investigated analyzing the MD trajectories of the complex with GeoMeasures. In this application, Geo-Measures was pivotal to easily compute and plot the RMSD, Rg, RMSF and FEL. Moreover, Geo-Measures was the method of choice for analyzing more flexible and scattered trajectories, due to the support for PCA, which is usually applied in these cases [96].

### 4.2. Enlighten2

Many environments for MD are available; however, the setup of programs for MD still requires detailed knowledge in the field. Enlighten2 [97], an easy-to-install interface to the Amber force-field [98], provides direct access to MD simulations. Protocols to set up MD simulations, which have already been tested and used in the previous version of Enlighten [99,100,101,102], together with the GUI, make MD accessible to all. Enlighten2 offers a GUI to several functionalities for molecule preparation (Figure 4g) and MD simulation set-up (Figure 4h). For instance, Enlighten2 implements Antechamber [103] and propka [104] to, respectively, compute ligands parametrization and to modify ligands and protein residues according to user-defined pH values. Furthermore, Enlighten2 supports the possibility to set up how to handle solvation and co-factors. When using Enlighten2, each step that modifies the protein’s properties is promptly visualized in PyMOL. Following molecule preparation, a dedicated object is created in PyMOL to store and visualize the solvent cap (Figure 4i). Moreover, the results are loaded in PyMOL as multiple separated objects. The interoperability of PyMOL and Enlighten2 is also emphasized in the tutorials provided on the Enlighten2 websites (https://enlighten2.github.io/tutorial1; accessed on 14 October 2022), where it is explained how to analyze mutant models of two enzyme-ligand complexes created with the PyMOL ‘Mutagenesis Wizard’. Providing easy access to MD simulations, also for non-expert users, it is now often a method of choice to support experimental data, as has been proposed for the engineering of glycan-binding proteins (GBP) [105].

**Figure 4 biomolecules-12-01764-f004:**
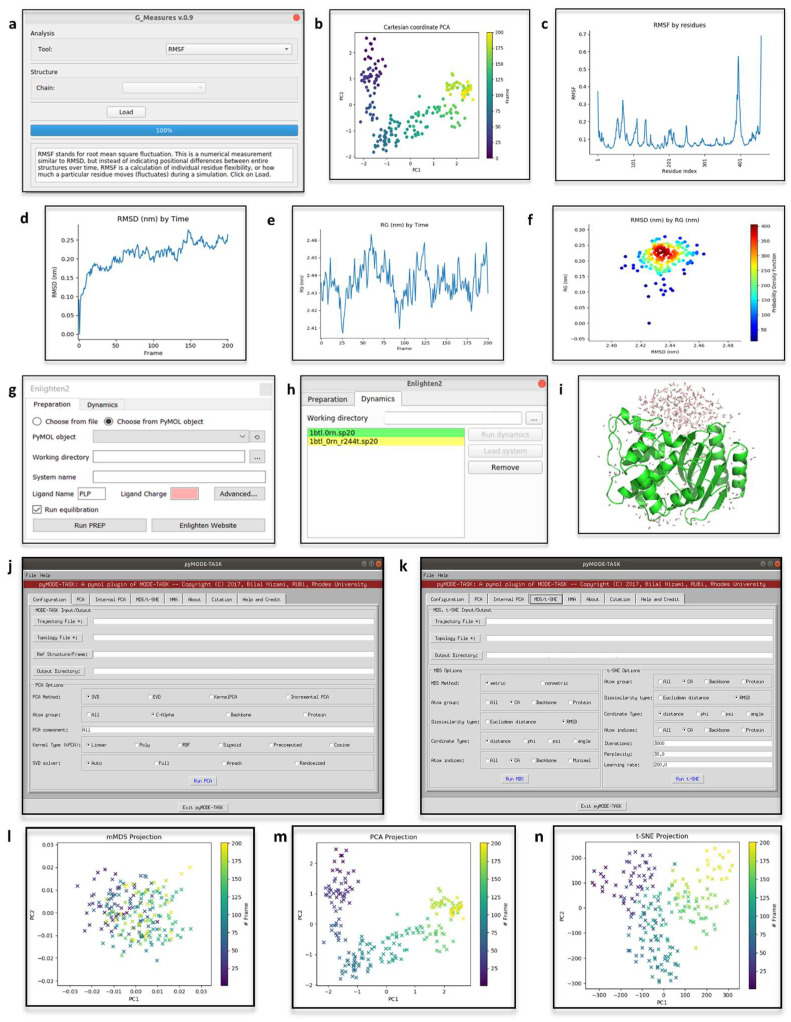
(**a**) Geo-Measures GUI (Ubuntu Linux OS) to run a Root Mean Square Fluctuations (RMSF) analysis on the MD-trajectory loaded in PyMOL. (**b**–**f**) Plots of the results of the analyses carried out from within Geo-Measures. In this example, the analyzed data were MD simulations of the multiple endocrine neoplasia type 1 (MEN1) [106]. On such trajectories, PCA (**b**), RMSF (**c**), RMSD (**d**), Rg (**e**) and PDF between RMSD and Rg values (**f**), were computed. (**g**) Enlighten2 GUI (Windows OS) to set-up the parameters for molecule preparation. (**h**) Enlighten2 GUI (Ubuntu Linux OS) to run the MD simulation. (**i**) Enlighten2 rendering of the solvent cap in PyMOL. In this example, the tutorial provided on the Enlighten2 website for beta-lactamase TEM-1 (PDB-ID: 1BTL; [107]) was followed. (**j**,**k**) Appearance of pyMODE-TASK (Ubuntu Linux OS) tabs for the setup of PCA, MDS and t-SNE analyses. (**l**–**n**) Plots provided by pyMODE-TASK to show the results of the PCA, MDS and t-SNE analyses carried out on MEN1 MD simulations data.

### 4.3. pyMODE-TASK

The analysis of PD data, either derived from MD simulations or coarse-grained Elastic Network Models (ENMs), is often a complex task; hence dimensionality reduction algorithms have taken hold. Even though many algorithms for deciphering biological relevant conclusions from PD have been developed, their availability as standalone command-line tools, or integration in commercial software, limits a free and easy application. In this context, a recently released tool, MODE-TASK [108] offers an array of implements for analyzing PD-derived data. MODE-TASK can be exploited to carry out the following analyses: (i) PCA (i.e., standard, kernel or incremental PCA); (ii) Normal Mode Analysis (NMA); (iii) Multidimensional Scaling (MDS); (iv) t-Distributed Stochastics, and (v) Neighbor Embedding (t-SNE). Furthermore, MODE-TASK incorporates a new algorithm, namely a coarse graining technique, which was validated for being less computationally expensive than those previously available. PyMOL has been chosen for hosting the GUI of MODE-TASK as a plugin, known as pyMODE-TASK (Figure 4j,k). Such integration makes the analysis of proteic large-scale motions easily accessible to PyMOL users. The pyMODE-TASK site provides introductory tutorials (https://pymode-task.readthedocs.io/en/latest/pyMODE-TASK_usage.html; accessed on 14 October 2022) on the use of this plugin, and its application in NMA and PCA of proteins (Figure 4l–n). PCA when applied to MD is pivotal to spot differences in the analysis of the same biological system in different states, as described for the destabilizing mutation of renin-angiotensinogen system [108], and the effect of phosphorylation on toll/interleukin-1 receptor domain-containing adapter protein (TIRAP)-mediated signaling [109]. In both works, pyMODE-TASK was exploited to compute and plot the principal components of the system.

## 5. Conclusions

It is clear that an interoperable use of software and tools can greatly facilitate the development of pipelines to approach CPA. Several algorithms with applications in this field have been developed accompanied by a command-line interface only, limiting their use to experts in computational biology. Moreover, given the computational capacity and precision, alongside existing ones, an increasing number of in-silico approaches will be available, placing the scientific community in front of a vast choice of alternatives. In this context, PyMOL is positioned as an integrative environment for bridging molecular visualization and CPA. Especially, PyMOL eligibility as a versatile platform, must be acknowledged in relation to accessing its framework through plugins. In addition, its most outstanding features, namely excellent image rendering, programmatic accessibility and inherent chemical editing tools, have contributed to making it an integral part of structural bioinformatics. This review highlights how the use of plugins that exploit and, at the same time, boost the use of PyMOL, can help create an environment to seamlessly carry out CPA.

## Figures and Tables

**Figure 1 biomolecules-12-01764-f001:**
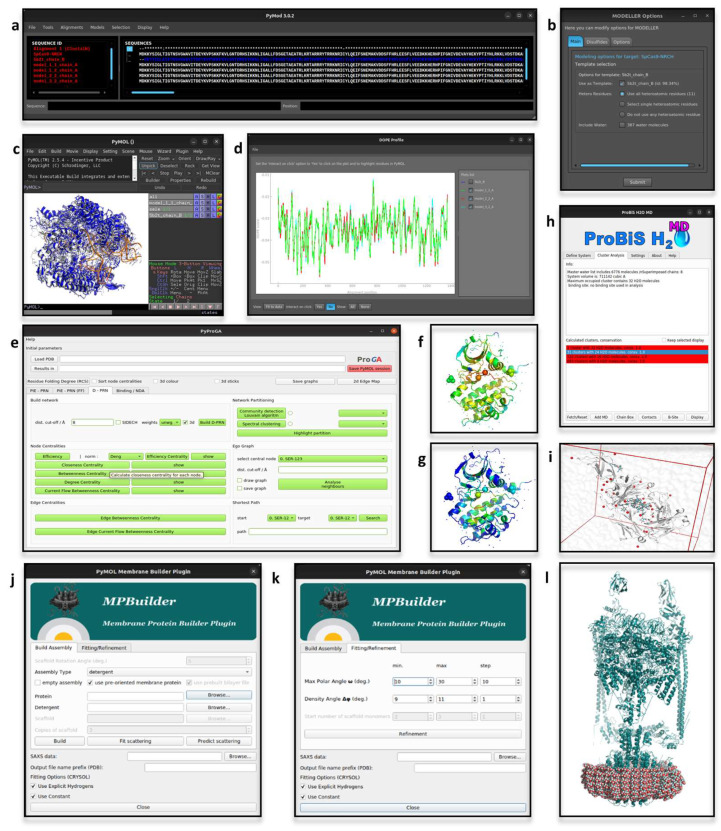
(**a**–**d**) Appearance of PyMod and PyMOL when running a template-based protein structure prediction. In this example, the ‘fasta’ file of the sequence to be modeled was directly opened in PyMod, while the template structure (PDB-ID: 5B2T [35]) was imported from PyMOL. The sequences of interest are visualized in the dedicated interactive window of PyMod, within which the analyses can be run (**a**). (**b**) PyMod window to set-up the parameters to run MODELLER [26] (e.g., how to consider heteroatoms, water molecules or disulfide bridges during calculations). (**c**) Visualization of the modeled structures in PyMOL workspace. (**d**) PyMod window for visualizing plots of the quality assessment [28]. (**e**) Appearance of the pyProGA window, from which the parameters for D-PRN analyses can be set-up. In this example, pyProGA was used to compute two different measurements on a protein (PDB-ID: 1OL5 [36]), which was directly loaded from PyMOL. (**f**,**g**) pyProGA processing of the proteins loaded in PyMOL according to the computed measures, which in this example were the ’Degree Centrality’ (**f**) and ’Betweenness Centrality’ (**g**). (**h**) Appearance of ProBiS H20 MD window for analyzing the results. Each identified cluster of conserved water molecules is identified in PyMOL as a red sphere (**i**). In this example the plugin was used to identify the conserved water molecules from molecular dynamics (MD) trajectories of a globular protein in water (the input files, a topology and a trajectory file, are provided by the ProBiS web-site (http://insilab.org/probis-h2o-md; accessed on 14 October 2022)). (**j**,**k**) Appearance of ‘Build Assembly’ and ‘Fitting/Refinement’ tabs of MPBuilder (Ubuntu Linux OS). (**l**) Visualization in PyMOL of MPBuilder output. The analysis reported here was carried out on the example files provided in MPBuilder development web-resource (https://github.com/emblsaxs/MPBuilder/tree/main/test_cases; accessed on 22 November 2022).

**Figure 2 biomolecules-12-01764-f002:**
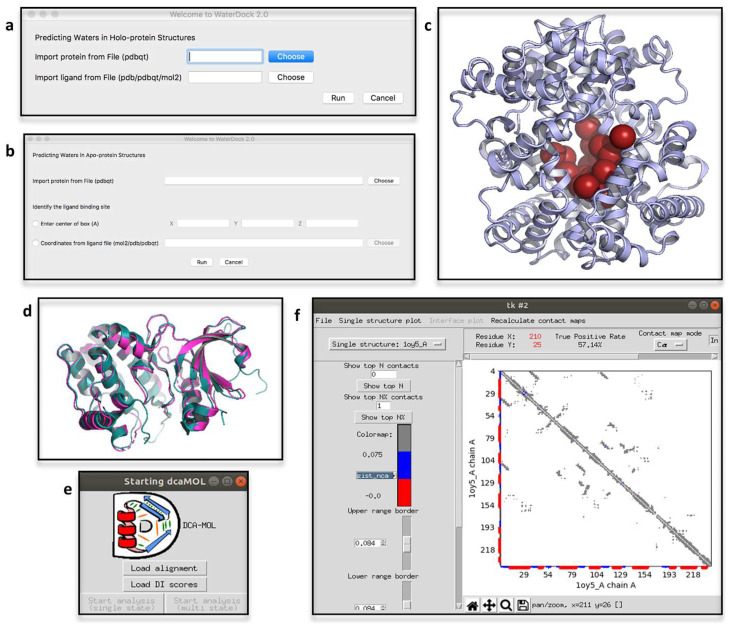
Waterdock 2.0 GUI (mac OS version) for the analysis of the position of water binding sites on holo-(**a**) or apo-(**b**) protein structures. Waterdock was tested with inputs available on the plugin development web-resource (https://github.com/bigginlab/WaterDock_pymol; accessed on 22 November 2022). (**c**) Waterdock rendering of the identified water molecules positions in PyMOL. (**d**) Example of PyMOL rendering of the pairwise structural alignment of two protein structures (i.e., Aurora-A kinase in two different conformations; PDB-ID: 1OL5 and 1OL6 [36]) obtained by iPBAvizu. (**e**) Starting window of DCA-MOL, from which alignments and DI-scores can be imported. (**f**) Example of a contact map obtained from DCA-MOL. Input files available on the plugin development web-resource.

**Table 1 biomolecules-12-01764-t001:** Summary of the available PyMOL plugins for CPA, released in the last years.

Name	Description	Release Date
DockingPie	A platform for molecular and consensus docking (PLI)	2022
PyMod	Environment for structural bioinformatics (PSSAs)	2021
pyProGA	Analysis of static protein residue networks (PSSAs)	2021
MPBuilder	Building and Refinement of Solubilized Membrane Proteins Against SAXS Data (PSSAs)	2021
PoseFilter	Filtering small molecule conformations ensemble (PLI)	2021
DRUGpy	Druggable hot spots identification (PLI)	2021
Geo-Measures	Analyses of protein structures ensemble (PD)	2020
Enlighten2	A platform for MD simulations (PD)	2020
ProBiS H2O MD	MD-based prediction of conserved water sites (PSSAs)	2020
iPBAVizu ^1^	Protein structure superposition approach (PSSAs)	2019
DCA-MOL ^1^	Analysis of Direct Evolutionary Couplings (PSSAs)	2019
pyMODE-TASK ^1^	Environment for MD trajectories analyses (PD)	2018
Waterdock 2.0	Water placement prediction (PSSAs)	2017
ProBiS H2O	Conserved water sites identification (PSSAs)	2017

^1^ Python 2 version of PyMOL is required (≤2.3). PSSAs, Protein Sequences and Structures Analyses; PLI, Protein-Ligand Interactions (PLI); PD, Protein Dynamics.

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
