# Peer review of "Boosting the Full Potential of PyMOL with Structural Biology Plugins"

_biomolecules, 2022, doi:10.3390/biom12121764_

Round 1
Reviewer 1 Report
The following summary discusses the review article, titled “Boosting the full potential of PyMOL with structural biology plugins,” written by Rosignoli S and Paiardini A.
This article discusses the PyMOL software platform that is utilized full biomolecules visualization and analysis. Rigorous amounts of details are discussed on the newly developed features available in the plugins. Specific content including the use case examples shared as part of the review. The authors have taken a great deal of time to explain how each of the plugins works and to highlights the core features of those plugins. This presents a concrete overview for users looking to explore these new plugins by also thinking about specific research questions that can be targeted within these. The paper focuses on 13 PyMOL plugins release in the past 6 years along with descriptions on how each of these plugins was designed; including their functionalities and capabilities. The review also highlights specific research questions that they will help to address. Detailed information is presented on CPA implementation of the plugins including tutorials and references for users to further explore during in silico experiments. Another important detail that is shared by the authors includes input data file types as well as example output from analysis. Operating system requirements and specific areas where users may connect to alternative workflows depending on their technological requirements are also discussed.
minor comments:
Image examples are shown for 8 out of the 13 plugins listed in Table 1. The authors should consider adding an additional figure panel (and accompanying figure legend) showing example output from the following 5 remaining plugins not currently displayed (DRUGpy, iPBAVizu, DCA-MOL, pyMODE-TASK and Waterdock 2.0). If these plugins are visually represented somewhere in their current figures then this should be clarified in the figure legend where appropriate.
Author Response
We wish to thank the referee for her/his appraisement and constructive criticism, and modified the revised manuscript accordingly, by adding additional figure panels for all the described plugins.
Reviewer 2 Report
The review article by Alessandro Paiardini and Serena rosignoli entitled ‘Boosting the full potential of Pymol with structural biology plugins’ gives valuable insights to the latest plugins and upgrades in pymol, a widely used molecular visualization and analysis tool. I believe this review articles expands the existing knowledge of python and I suggest this can be published without any further revision/corrections.
Author Response
We really appreciated the assessment and appraisal of the referee.